# Perspectives of health workers on the referral of women with obstetric complications: a qualitative study in rural Sierra Leone

Ryan Proos,[1,2] Hanna Mathéron,[1] Jonathan Vas Nunes,[1] Abdul Falama,[3] Patricia Sery Kamal,[3] Martin Peter Grobusch  ,[1,4] Thomas van den Akker[2,5]

[1]Masanga Medical Research Unit, Masanga, Sierra Leone
[2]Obstetrics and Gynaecology Department, Leiden University Medical Center, Leiden, The Netherlands
[3]Tonkolili District Health Management Team, Magburaka, Sierra Leone
[4]Center of Tropical Medicine and Travel Medicine, Department of Infectious Diseases, Amsterdam University Medical Centres, Duivendrecht, The Netherlands
[5]Athena Institute, VU Amsterdam, Amsterdam, The Netherlands

**Correspondence to**
Professor Martin Peter Grobusch;
m.p.grobusch@amsterdamumc.nl

## ABSTRACT

**Objectives** Sierra Leone has one of the highest maternal mortality ratios in the world. Timely and well-coordinated referrals are necessary to reduce delays in providing adequate care for women with obstetric complications. This study describes factors affecting timely and adequate referral of women with obstetric complications in rural areas of Sierra Leone as viewed by health workers in rural health facilities.

**Design** Qualitative research with semi-structured interviews using open-ended questions. Data were analysed by systematic text condensation.

**Setting** Interviews were held in nine peripheral health units in rural Sierra Leone.

**Participants** 19 health workers including nurses, midwives and clinical health officers participated in nine interviews.

**Results** From the interviews, four major themes describing possible factors of delay in referral of women in need of emergency obstetric care emerged: (1) communication between healthcare workers; (2) underlying influences on decision-making; (3) women's compliance to referral and (4) logistic constraints.

Several factors in rural Sierra Leone are perceived to complicate timely and adequate referral of women in need of emergency obstetric care. Notable among these factors are fear among women for being referred and fear among healthcare workers for having maternal deaths or severe obstetric complications occurring at their own facilities. Furthermore, decision-making of healthcare workers whether to refer a woman or not is negatively influenced by a hierarchical culture with high power distance between healthcare workers.

**Conclusion** Factors identified that complicate timely and adequate referral of women in need of emergency obstetric care must be considered in efforts to reduce maternal mortality. Possible interventions that may reduce delay in referral include increased communication by mobile phones between health workers for advice and feedback regarding referrals, involvement of influential stakeholders to increase women's compliance to referral, and consistent use of standardised management protocols.

---

### Strengths and limitations of this study

► A strength of this study is the purposive and homogenous sampling used for the selection of peripheral health units and health workers for the interviews, which is representative of health facilities and health workers in rural Sierra Leone.
► A strength of this study is the use of open-ended questions alongside an interview guide, ensuring specific topics were discussed while allowing participants to introduce and discuss additional topics.
► A limitation of this study is that the data collected using semi-structured interviews with health workers were not triangulated with data from interviews with women and relatives.

---

### BACKGROUND

Sierra Leone has one of the highest maternal mortality ratios (MMR) in the world. According to the Sierra Leone Demographic Health survey in 2013, the MMR was 1165 per 100 000 live births.[1] The need to reduce this extremely high ratio is evident. In order to work towards meeting the United Nations Sustainable Development Goal target of an MMR below 70 per 100 000,[2] the Ministry of Health and Sanitation of Sierra Leone launched the Reproductive, Maternal, Newborn, Child and Adolescent Health Strategy in 2017, aiming to reduce the MMR of 1165–650 by 2021.[3]

An underlying factor of the high MMR in Sierra Leone is the persistent low rate of facility-based births.[1] Country-wide in 2013, only slightly more than half of the women gave birth in a health facility; 49.7% of women in rural areas versus 68.1% in urban areas.[1] The West-African Ebola outbreak from 2013 to 2016 led to a further reduction in facility-based births as a consequence of reduced possibilities and fear to access the health system during the crisis.[4 5] Pregnant women in Sierra Leone face many barriers to facility-based birth, including long

| Table 1 | Maternity services purportedly provided at each level of health facility |
|---|---|
| Maternal and Child Health Post (MCHP) | ► Antenatal care.<br> – Nutritional supplementation in pregnancy (eg, iron, folic acid and multivitamins).<br> – Risk selection and ensuing referral.<br> – Malaria intermittent preventive treatment.<br>► Intrapartum and postpartum care.<br> – Monitoring of labour by using the partograph.<br> – Cord clamping.<br> – Active management of the third stage of labour.<br>► Postnatal care.<br> – Clinical assessment of the neonate (eg, fever, convulsions, feeding).<br> – Exclusive breast feeding recommendation.<br> – Cord care.<br> – Clinical assessment of mother (eg, temperature, blood pressure, bleeding).<br> – Family planning counselling. |
| Community Health Post | ► MCHP services (see above). |
| Community Health Centre | ► MCHP services (see above).<br> Plus<br>► Maternal anaemia and urine sediment assessment.<br>► BEmONC services (see Online Supplemental File 2). |
| District hospital | ► MCHP services (see above).<br> Plus<br>► Maternal anaemia, urine, HIV, malaria and tuberculosis assessment.<br>► Ultrasound scan.<br>► CEmONC services (see Online Supplemental File 2). |

BEmONC, Basic Emergency Obstetric and Neonatal Care; CEmONC, Comprehensive Emergency Obstetric and Neonatal Care.

distances, inability to afford costs of transport and healthcare and lack of trust in health facilities.[6] These factors contribute to a phase I delay in deciding to seek healthcare in case of an emergency obstetric complication, as described in the three-phase delay model by Thaddeus and Maine.[7] Even after having decided to seek healthcare, women are often confronted with delays in phase II (transport delay), and phase III (delay in diagnosis and treatment at the facility).[8] Shortly before this study was conducted, a new ambulance system was implemented in Sierra Leone, possibly reducing transport delays. Timely and well-coordinated referrals are necessary to further reduce delay in receiving adequate obstetric care for women with obstetric complications. Currently, there is no literature available analysing aspects of reasons for delay of care within the obstetric referral system in rural Sierra Leone.

This study describes factors affecting timely and adequate referral of women with obstetric complications in rural areas of Sierra Leone through the perspectives of health workers in rural health facilities. This data will provide better understanding of challenges within the health system faced by women who are referred with emergency obstetric complications in rural Sierra Leone.

## METHODS
### Study design and setting
This qualitative study using semi-structured interviews and following the Consolidated Criteria for Reporting Qualitative Research checklist[9] (Online Supplemental File 1) was conducted between the 1st of September 2018 and the 15th of March 2019 in Tonkolili District. This district is located in the Northern Province of Sierra Leone and sub-divided

into 11 chiefdoms. Three hospitals with Comprehensive Emergency Obstetric and Neonatal Care (CEmONC) are located in Tonkolili District. Basic Emergency Obstetric and Neonatal Care (BEmONC) and CEmONC are services fundamental to provide adequate healthcare during pregnancy and childbirth. The signal functions of BEmONC and CEmONC centres are summarised in Online Supplemental File 2. Three chiefdoms in the north of Tonkolili District, Kafe Simiria, Kalansogoia and Sambaya Bendugu, with a combined population of 113 521 (2018), are served by two CEmONC centres, Magburaka Government Hospital and Masanga Hospital, an non-governmental organisation (NGO)-supported government hospital. Besides the two CEmONC centres, these chiefdoms are served by fifteen peripheral health units (PHUs), including four BEmONC centres. The maternity services provided at each facility level in these chiefdoms, according to the Ministry of Health and Sanitation, are summarised in table 1.

However, not all facilities designated to provide BEmONC and CEmONC in rural Sierra Leone are able to provide the full range of signal functions.[10 11] Therefore, several BEmONC and CEmONC signal functions might not actually be provided in practice. Referral occurs both in consecutive order starting from Maternal and Child Health Posts (MCHPs) as well as between lower level health facilities and district hospitals. Sierra Leone has national protocols for emergency obstetric care, including referral indications.[12]

Tonkolili District was conveniently selected, since the district is located in rural Sierra Leone and the catchment area of the hospital where three of the authors are employed. The chiefdoms in northern Tonkolili District were selected,

**Table 2** Facilities where interviews were conducted

| Town | Level | Distance to district hospital (km) | Travel time to district hospital* (min) |
|---|---|---|---|
| Chiefdom Kafe Simiria | | | |
| Mabontor | CHC | 18.9 | 40 |
| Masumbrie | CHC | 21.5 | 40 |
| Makontande | MCHP | 28.9 | 50 |
| Chiefdom Kalansogoia | | | |
| Bumbuna | CHC | 42.6 | 70 |
| Kamasaypana | MCHP | 50.0 | 100 |
| Kemedugu | MCHP | 58.5 | 110 |
| Chiefdom Sambaya Bendugu | | | |
| Bendugu | CHC | 81.4 | 150 |
| Kunya | CHP | 92.3 | 180 |
| Dankawalia | MCHP | 71.9 | 135 |

*Travel time by motorbike during dry season (November–May). Travel time during rainy season (June–October) will be substantially longer. Travel time by ambulance will be shorter. Roads were unpaved.
CHC, Community Health Centre; MCHP, Maternal and Child Health Post.

since these three combined comprise the catchment area for emergency obstetric complications belonging to Masanga Hospital, for reasons of geography such as impassable rivers and mountains and accessibility by road.

The qualitative data used in this study were collected using nine semi-structured interviews conducted by RP (Master of Medicine student, male, first author) between November 2018 and January 2019. Purposive sampling was used to ensure variation between selected facilities where interviews were conducted. Selection criteria were that facilities were located in different chiefdoms, provided different levels of care, and had varying accessibilities of the nearest district hospital (table 2).

Selected facilities were approached either by telephone calls to the in-charge health worker or by face-to-face visits to that health facility. All health workers working at the selected facilities at the time of the interview were invited to participate in the interview. Thus, homogenous sampling was used as each interview was conducted with health workers of different cadres currently working in the same health centre. The number of participants per interview ranged from one to four. In total, 19 health workers participated in the nine interviews. All participants were

explained the relevance and goals of the research. No facility or individual health worker refused participation. Respective health worker cadres and competencies of the participants are summarised in table 3.

The interview guide (Online Supplemental File 3) was used as a framework of themes to be discussed during the interviews. The guide was initially developed using themes described by Thaddeus and Maine,[7] previous literature concerning pregnancy and childbirth in Sierra Leone,[6 13] and preliminary discussions with stakeholders such as medical officers, community health officers (CHOs), midwives and logistical officers employed at district referral hospitals. The guide was piloted in one PHU with two health workers. The interviews were held inside the respective health facilities where only participants and interviewer were present. The interviews were conducted in English, using open-ended questions, and lasted between 30 and 60 min each. Data were collected using audio recording. Repeat interviews were not carried out and transcripts were not returned to participants for comments and correction for logistical reasons: there was no funding or practical possibility to revisit these widely spread-out facilities. The

**Table 3** Cadres and competencies of respondents

| Health worker (number interviewed) | Competencies |
|---|---|
| Maternal and Child Health aid (10) | 2 years training. Competent in basic obstetric care. |
| State Enrolled Clinical Health Nurse (SECHN) (2) | 2.5 years training. Competent in basic obstetric care. |
| Community Health Assistant (3) | 2 years theoretical +1 year practical training. Competent in basic obstetric care. |
| Community Health Officer (2) | 3 years theoretical +1 year practical training. Competent in basic obstetric care. No training in emergency obstetric care. |
| Midwife (2) | SECHN training +1.5 year-midwifery training. Competent in emergency obstetric care including oxytocin administration, manual placenta removal, newborn resuscitation, first treatment for (pre)eclampsia and antibiotic administration. No training in remaining signal functions. |

original study protocol (Online Supplemental File 4) was followed throughout the study.

### Study subject and public involvement
The District Health Management Team was involved in the design, conduct, reporting and dissemination planning of our research. All participants of the interviews were informed on relevance and goals of the study.

### Analysis
All audio recordings of interviews were transcribed verbatim by RP using Express Scribe Transcription Software (NCH Software, Greenwood Village, Colorado, USA). Content analysis was performed by RP through systematic text condensation as described by Malterud.[14] After all interviews were transcribed verbatim, all transcripts were read multiple times to establish an overview of the data. Preliminary themes were identified based on this overview. Hereafter, all transcripts were once again reviewed line by line, to identify 'meaning units', that is, text fragments containing some information about the research question. These meaning units were marked with a code: a label that connects related meaning units into a code group. These code groups were elaborated from the themes from the first step of the analysis. Hereafter, the meaning units in each code group were connected to form a condensate. Finally, these condensates were synthesised to accurately reflect the original quotes and some original quotes were included in the text to further illustrate the data. Data analysis was performed manually. HM was involved in the whole process of data analysis and gave feedback on the identification of the preliminary themes, the systematic coding and categorisation of quotes, and the writing of condensates of every theme based on the quotes.

### RESULTS
From these interviews, four major themes describing possible factors of delay in referral of women in need of emergency obstetric care emerged: (1) communication between healthcare workers; (2) underlying influences on decision making; (3) women's compliance to referral and (4) logistic constraints.

### Importance of communication between staff of different health centres
#### Giving and receiving advice surrounding referrals
Most health workers mentioned the necessity of asking for advice when having to decide whether to continue management or refer the woman to a higher-level health centre. One community health assistant (CHA) explains:

> I am not saying 100% I know what I am doing. I know myself. I learn, I can just know my own area and then there are people who know better. I am just a community health officer, assistant in fact. (male CHA, MCHP)

Advice was often asked for and given in mobile phone conversations. However, sometimes higher cadre health workers travelled to health facilities to review women themselves before advice was given.

> Sometimes he will come and he will review the patient and tell us to send the patient, so we call the ambulance. (female MCH aid, CHP)

Those asked for advice include CHOs, midwives, the District Health Sisters (supervising midwives, members of the Tonkolili District Health Management Team), the head of a maternity ward or a medical officer at a district hospital. However, one maternal and child health aid (MCH aid) mentioned that if she recognised a woman requiring urgent referral to a higher-level facility, she would not delay by first calling for advice but rather directly refer the woman to the Community Health Centre (CHC). She would not, however, directly refer the woman to a district hospital or inform the CHC that this woman likely needed onward referral, since she believed that this decision had to be made by CHC staff.

Besides needing advice on whether or not to refer the woman to a higher-level facility, participants also mentioned a need for advice over the phone regarding clinical management while waiting for the ambulance to arrive, since this could take up to several hours.

### Feedback after referral
Many health workers indicated that they were interested in the clinical course after a woman had been referred to the district hospital, as illustrated by one CHO:

> We are highly interested in feedback, because they are lives and when we call on you people to rescue, then we have interest over them. (male CHA, CHC)

Health workers in lower level facilities expressed a specific interest in knowing the clinical management including decisions on mode of birth at the district hospital. These health workers were often approached by relatives of the referred woman requesting updates on her clinical condition and outcome. Health workers regularly marked their phone numbers on the referral notes in order to receive feedback. However, many respondents reported never receiving a response from district hospital staff. Instead, they felt forced to call the district hospital themselves, and indicated this comprised a communication barrier, since it required them to spend their own mobile phone credits.

> But they don't give us the feedback for us to know if it was a vacuum delivery or what. I have no response … (female MCH aid, MCHP)

Similarly, health workers reported never receiving discharge notes with follow-up information after the woman had been discharged from the district hospital. They generally relied on the information the woman could give them verbally.

## Underlying influences on decision-making
### Referral, perceived as the safest option for health workers

The necessity of referral, as expressed by one health worker, was often to avoid complications and maternal death occurring at their own facility.

> If a maternal death is here, we are going to suffer. (female SECHN, MCHP)

Often, a referred woman was described as 'not my case'. Another health worker mentioned that when a woman was referred to a CHC, it was up to that facility to manage the woman with the complication and decide what to do. Such transfer of responsibility after referral was further illustrated by a story recounted by an MCH aid about a woman she had recently referred:

> Yes, I delivered her. Male baby. Fresh still birth. So it is not my problem, because I have already referred her. (female MCH aid, MCHP)

### Endangering behaviour by women requiring referral

Some women did not want to be referred and health workers were under a lot of pressure from women and relatives while making a referral decision. Women and relatives were at times perceived not to tell the truth when questioned about history, since they wanted to prevent referral. One health worker mentioned that he sometimes heard rumours in the community that the point in time a woman and her relatives indicated as when onset of symptoms occurred was not always correct.

> Nothing of the time stated was really factual, it was not the time, the actual time. (male CHA, MCHP)

Most respondents described that when a woman and her relatives were told that she needed to be referred, they started begging health workers not to refer her but to continue clinical management at the same facility. Women and relatives would try to convince them that they, as health workers, would be able to manage the woman with the complication without referral. One State Enrolled Clinical Health Nurse (SECHN) stated:

> They will want us to do everything while we don't have that ability. (female SECHN, CHC)

A CHA voiced his frustration at the women's and relatives' behaviour and stated that it endangered his own work.

### Women's compliance with referral
#### Influential stakeholders involved to improve compliance

Referral to a district hospital was perceived to come with many fears and worries for a pregnant woman. Examples mentioned by health workers included fear of undergoing surgery, viral haemorrhagic fever (Ebola or Lassa virus) infection, blood donation, male health workers and an unfamiliar environment in terms of language and people. Such fears contributed to women returning home instead of travelling to the health centre they were referred to. Respondents identified three influential stakeholders who may potentially reduce fears around referral. The first stakeholder was the chief of the village or town, whose advice and instruction were of substantial influence on the women's and relatives' referral compliance.

> My chief is not staying that far from me. So sometimes I call him when there is a case, when there is need for him to come here, as stakeholder, that he can talk to this person. (male CHA, MCHP)

The second group of stakeholders, recognised by health workers as similarly influential, consisted of relatives in Freetown, the capital of Sierra Leone. One SECHN stated:

> They will always listen to their relatives out there. (female SECHN, MCHP)

These relatives in Freetown were contacted and requested to attempt to convince the woman of the necessity of the referral and to adhere to the referral instructions. Finally, according to the health workers, women who had previously been referred to a district hospital and returned safely had a positive influence on a woman's perceptions regarding referrals to district hospitals.

### Logistical constraints
#### Medicine shortage as a burden on the referral system

The logistical constraints of dealing with stockouts of medication in PHUs comprised an additional burden on the referral system. Many health workers, especially those working in CHCs, reported struggling with medication shortages. Injectable antibiotics were often mentioned as insufficient for the purposed term.

> But medication, logistics, is much more paramount. We need IV fluids, we need drugs. (male CHA, CHC)

When medication stocks had been exhausted, the health worker was faced with two options. The first option was to request money from the woman to purchase medication at a local pharmacy. One health worker described the friction this created with the Free Health Care Scheme for pregnant and lactating women, since it was by law illegal to request money from these women.

> I am not going to ask her to pay for the service I am rendering but just to provide the drug. But me that is punishable crime, I cannot … Then I be in fault. (male CHA, CHC)

However, sending a woman directly to the pharmacy to buy medication themselves was not safe according to several health workers. They expressed their distrust in local pharmacists as they suspected them of not being properly trained and sometimes giving the wrong medication as well as administering injectable drugs themselves against regulations.

Health workers were therefore often forced to resort to referral to a district hospital in order for a woman to access the correct medication. The CHA voiced his desperation:

So what would you do? You just have to refer. (male CHA, CHC)

He also expressed his worry about the reaction from the district hospital after receiving such referrals. He feared that the district hospital would doubt the competency of the health workers at the PHU referring a woman who could potentially be managed at their own facility. Finally, some respondents pointed out that referring such a woman exposed them to additional adverse outcomes such as the relatives falling back on traditional medicine or going to a local pharmacy, since this was cheaper than paying for referral transportation.

### Inadequate ambulance availability

Tonkolili District has a limited number of ambulances available for the transport of women with emergency obstetric complications. Many health workers reported about the fact that when they call for an ambulance, they are told that the ambulance has broken down or to wait since the ambulance is on its way to a different, sometimes very distant PHU.

If you call the ambulance, at times the ambulance takes three to four hours before arriving here. (male CHO, CHC)

It was also noted that it sometimes takes a long time before the ambulance team, comprising a driver and a nurse, is mobilised at the district hospital and the ambulance is finally under way. One CHA summarised the problem as:

So sometimes it's very difficult; the time the ambulance is here, the patient is seriously in a critical condition. (male CHA, CHC)

Another problem reported was the road accessibility of certain PHUs. Some of these PHUs can only be reached by motorbike and on foot. Accessibility is worse during rainy season. Ambulance transport during the rainy season was even stated to be not possible at all for several PHUs.

### DISCUSSION

This study highlights several aspects of the obstetric referral system in rural Sierra Leone, which require attention in order to provide timely and adequate management of women with emergency obstetric complications.

### Importance of communication between staff of different health centres

The importance of communication between health centres to achieve an effective referral system was widely acknowledged by participating providers. Healthcare workers generally concurred with each other on the advantages of receiving advice on whether to continue management, or to refer a woman instead. However, our findings concerning the practice of waiting for a higher-cadre health worker from a different health facility to arrive and personally examine the woman before

advising on a referral decision, are in disagreement with protocol and would increase type 2 delay. This delay can be largely abated by adequate use of mobile phones to communicate with higher-cadre health workers for advice regarding referral decisions,[15] as well as further education for MCH aids who will not need a second opinion anymore. Furthermore, the reluctance of lower-cadre health workers in referring women directly to a district hospital and thereby bypassing the higher-cadre health worker's judgement as well as the reluctance of referring women to a CHC with the advice to further refer to a district hospital reveal potentially harmful hierarchy between health workers. From Tanzania, Ueno et al[16] reported a similar atmosphere of hierarchy and lack of cooperation between different cadres of health workers and levels of health facilities as a challenge to EmOC service delivery.

Our findings also imply that following referral, district hospitals need to take initiative in providing health workers in PHUs with feedback in order to improve and encourage future referrals and follow-up management of the woman after discharge from the district hospital. Studies in Ghana, Burundi and Northern Uganda reported similar demands for feedback after referral.[17–19] Multiple other studies have described related gaps in communication surrounding obstetric referrals[20 21] and have specified the critical role of communication in an effective referral system.[21–24] Improvement in communication between health facilities and health workers is a necessary first step towards improving the referral system in northern Tonkolili District.

### Underlying influences on decision-making

Our findings point at a mindset of some health workers regarding obstetric referrals that has potential adverse effects on timely management of women with emergency obstetric complications. Maternal mortality and morbidity were seen as tragic events for women and relatives, but also alarming for themselves as health workers. Referral and the ensuing transfer of responsibility was regarded as an option to prevent themselves from being blamed in case of a complication. It is hypothesised that this mind-set is an adverse result of the increased awareness of, and attention to, the high maternal mortality and morbidity rates in Sierra Leone. Obstetric audits have been proven to be an effective method of reducing maternal mortality and morbidity,[25–27] but a negative impact on work satisfaction and motivation have also been reported.[27 28] However, our findings provide limited evidence and further research in rural Sierra Leone is essential to accurately analyse this information.

Another underlying influence which became apparent throughout the interviews was the persuasiveness of women and relatives who did not want to be referred to a different health facility. Such persuasiveness has potential to delay the referral decision made by a health worker and thus results in phase I and phase II delays. Also, it may lead to over-confidence of lower-cadre health

workers respecting their ability in managing women with obstetric complications, as has been previously reported in Sierra Leone by Theuring *et al*.[13]

### Improving women's compliance with referral

Fears experienced by women, as reported by health workers, to accept referral to a district hospital such as fear of operations, blood donation, male health workers conducting births, and a new environment were largely similar to those previously reported by multiple studies in similar settings.[17 19 29 30] A lingering fear of Ebola in district hospitals in post-Ebola regions including Sierra Leone has been described in several other studies.[13 31 32] Three groups of influential stakeholders were identified: local village chiefs, relatives in the capital, Freetown; and women who have previously been referred to a district hospital and have returned safely. Consulting these stakeholders in case of referral refusal will potentially increase women's compliance. These stakeholders are understood to be similarly influential in all rural areas of Sierra Leone.

### Logistical constraints

The influence of medication shortage in PHUs on the effectiveness of the referral system becomes apparent through our findings. The combined effect of the Free Health Care Scheme[33] and the distrust in local pharmacies forces health workers into avoidable referrals to district hospitals. In turn, these referrals lead women to a choice of options from traditional medication or buying medication from a local pharmacy. The shortage of medication in PHUs is a burden on and complicating factor of the referral system in northern Tonkolili District; however, adequate availability of medication should not avert mandatory referrals of women with obstetric complications, which require management in district hospitals.

The shortage of ambulances for transport of women from lower-level health facilities to district hospitals is a commonly reported contributor to phase II delay in sub-Saharan Africa.[17–19 34 35] Our findings show that this barrier to access to adequate emergency obstetric care is also present in rural Sierra Leone. The inability of ambulances to reach certain PHUs, due to arduous terrain such as steep hills and river crossings and poor road conditions, worsened by seasonal rains, displays the poor infrastructure of rural Sierra Leone.

In February 2019, after our data collection was completed, the Ministry of Health and Sanitation of Sierra Leone and the NGO Doctors with Africa CUAMM launched the National Emergency Medical Service (NEMS), which provides free-of-charge ambulance service in all of Sierra Leone. The implementation of the NEMS allows for imperative reduction in delay in access to adequate emergency obstetric care in district hospitals. This reduction in delay is possible as the number of functioning ambulances is increased, and as the travel time is decreased as soon as the ambulances are not only stationed in district hospitals any longer, but also in PHUs.

### Strengths and limitations

A strength of this study is the selection of PHUs and health workers for the interviews, which is representative of health facilities and health workers in rural Sierra Leone. Therefore, interventions targeting the obstetric referral system in other areas of rural Sierra Leone can be supported by our findings. Another strength is the use of open-ended questions during the interviews, which allowed the participants to express their own experiences and feelings. Additionally, as this study was conducted before the implementation of the NEMS, this study allows for a follow-up study analysing the effect of the NEMS. The authors hypothesise that only the themes 'communication' and 'logistical constraints' have been affected by the NEMS.

A limitation of this study was that the data collected using semi-structured interviews with health workers were not triangulated with data from women and their relatives. However, results from previously conducted research concerning the perspectives of women and their relatives have been discussed to further validate our data. Additionally, a description of the major reasons of referral would have been of added value to our manuscript. Finally, the interviews were conducted in English while the participants were more familiar with Krio.

### CONCLUSION

In the perspectives of healthcare workers, delay in access to adequate emergency obstetric care is caused by lack of communication between health workers at different facilities, lack of involvement of influential stakeholders, medication shortage and lack of ambulance services. Of note, fear among women and their relatives for them to be referred is another cause of delay. Furthermore, the decision-making of healthcare workers concerning referral is negatively influenced by an atmosphere of hierarchy and fear of having maternal deaths and other severe complications at their facility.

Interventions that may reduce delay in access to adequate emergency obstetric care include communication by mobile phones for advice regarding referral decisions and for feedback after a referral decision has been made. Involvement of influential stakeholders to increase women's compliance to referral is an additional intervention that may be considered. Additionally, consistent use of a standardised management protocol at the different levels of health facilities may reduce delay in access to emergency obstetric care.

This study highlights factors that may complicate timely and adequate referral of women in need of emergency obstetric care. As this delay is an underlying cause of the high MMR in rural Sierra Leone, these potential sources and causes of delay must be considered in efforts to reduce maternal mortality.

**Acknowledgements** The authors wish to acknowledge all the staff in healthcare facilities that participated in the study.

**Contributors** RP carried out the field work, analysed the data and wrote the first draft of the paper. HM and JVN conceived the study and contributed to phrasing the study question, data interpretation and writing of the paper. AF, PSK and MPG contributed to data interpretation and writing of the paper. TvdA oversaw the conduct of the study and contributed to data interpretation and writing of the paper. All authors have contributed to the writing of, and approved the final version of the paper.

**Funding** The authors have not declared a specific grant for this research from any funding agency in the public, commercial or not-for-profit sectors.

**Competing interests** None declared.

**Patient consent for publication** Obtained.

**Ethics approval** The study proposal was endorsed by the Masanga Medical Research Unit Scientific Review Committee. Ethical approval was obtained from the Sierra Leone Ethics and Scientific Review Committee. Permission to conduct the study in Tonkolili District was obtained from the District Health Management Team. Written informed consent was obtained from interview participants.

**Provenance and peer review** Not commissioned; externally peer reviewed.

**Data availability statement** Data are available upon reasonable request. The datasets used and/or analysed during the current study are available from the corresponding author on reasonable request.

**ORCID iD**
Martin Peter Grobusch http://orcid.org/0000-0002-0046-1099

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
