## [Reviewer comments · BMJ Open]

ARTICLE DETAILS

TITLE (PROVISIONAL)	Perspectives of health workers on the referral of women with obstetric complications: a qualitative study in rural Sierra Leone
AUTHORS	Proos, Ryan; Matheron, Hanna; Vas Nunes, Jonathan; Falama, Abdul; Sery Kamal, Patricia; Grobusch, Martin; van den Akker, Thomas

VERSION 1 – REVIEW

REVIEWER	Ibrahim Kabo Alhassan Palladium International Limited, Integrated Health Program (IHP) Nigeria
REVIEW RETURNED	29-Jul-2020

GENERAL COMMENTS	General Comments I commend the authors for the good work they have done. The study has identified critical issues and gaps to provision of timely and adequate referral for women with emergency obstetric complications in rural Sierra Leone. The topic is relevant and addresses an important problem that faces not only Sierra Leone but also other developing countries. Section specific remarks Title: Clearly described the article Abstract: Objectives: It would be better if the objectives explain the purpose of conducting the study beyond just describing the perspectives of health workers. I would suggest for consideration the objective to be “Investigate factors affecting timely and adequate referral of women with obstetric complications in rural areas of Sierra Leone” Results: Line 41. Furthermore, decision-making of I suggest to rephrase the sentence for clarity ARTICLE SUMMARY Line 51 Strengths and limitations of this study Other limitations of the study: 1. The study did not consider getting perspective of women and relatives on “women’s compliance to referral” Background: A brief information that describes the current referral system in Sierra Leone could be of added value. Why the choice of Tonkolili District in Sierra Leone? Data Collection Line 123 Data collection section as described in the manuscript should be moved to Study Design and Setting section line 95 In addition, under the methodology 1. There is no mention of research subjects – only engagement with district health management team was mentioned.2. Authors may consider listening the inclusion and exclusion criteria since is a purposive sampling3. The authors mentioned that a homogenous sampling was used
---

	for the health workers without explaining how it was done. The Data Collection Section should basically describe how the data was collected as per design, needs to adjust the data collection instruments, challenges encountered etc Line 136 Table #3 under competencies of “Midwife 2”. I would be of added value to list the remaining 3 signal functions and whether she could perform them or not Results: The Authors would have included the major/common reasons for referring women to higher level health facility Line 232 “Women’s compliance with referral” The Authors would have considered getting perspective of women and relatives on to balance/validate health workers” perspective Strengths and Limitations Another limitation is that the study did not consider getting perspective of women and relatives on “women’s compliance to referral” Conclusion The claims are supported by the results. Other interventions that may reduce delay in access to adequate emergency obstetric care can based on the findings also include competency based trainings, use of standardized management protocols at different levels, improving referral system with 2-way referral, clinical mentoring.
--	---

REVIEWER	Antonia Arnaert McGill University, Ingram School of Nursing, Canada
REVIEW RETURNED	10-Aug-2020

GENERAL COMMENTS	Paper: Perceptions of health workers on the referral of women with obstetric complications: A qualitative study in rural Sierra Leone. This is an interesting topic to explore; however, the paper needs Abstract:  • In the title the authors are talking about “perceptions” and in the abstract they indicate “perspectives”? • What is “systematic text condensation”? • More information should be provided on those 19 health workers. Nurses? Midwives? • The results indicate 4 themes but what is the relationships between those themes and the factors that complicate timely and adequate referral? • The abstract should be improved Strengths and Limitations  • Is it a limitation that the interviews were not triangulated with quantitative data? Why? • A limited number of participants? The authors interviewed 19 participants. Why is this a limitation? Background  • No literature available? What about evidence in other countries? Method  • Why is it important to say that the District Health Management Team was involved in the design, conduct, reporting and dissemination planning of our research? • In the paper it says 9 semi-structured interviews (instead of 19). What is the number of participants?
---

	 • Is the Master of Medicine student a co-author? Perhaps indicate that the interview by 1st or ... co-author? • How were those facilities selected? Based on the number of high-risk pregnancies? What is the % of at-risk mothers in those districts? • A purposive sampling was used to select facilities and a homogenous sampling was used to select healthcare workers? Unclear/ • 19 Healthcare workers participated in 9 interviews? Why? How was this done? Focus groups? • How were those healthcare workers selected? • In the table it says: "health worker (number interviewed)". No number is provided. • What about trustworthiness of data? Results  • There is no introduction to the results? • It would be good to describe the roles of those health workers in case of a referral. • The nuances of those different health workers must be better described in the results. • Quotes are not rich descriptions and not always meaningful to the text. • Why using references in results? Under logistical constraints Discussion: No discussion provided
--	---

REVIEWER	Adelaide Lusambili Aga Khan University Kenya
REVIEW RETURNED	21-Aug-2020

GENERAL COMMENTS	 1. Line 2: COREQ - spell this acronym and all other acronyms in the paper first time. 2. Abstract Remove HCWs from settings and leave it under the heading "participants". 3. Line 33-60. Remove bullet points. Write as a paragraph. You have the same information from line 374. I suggest, you delete this information from line 33 -60 and move to line 374. 4. Line 120, remove heading and merge line 121-122 with the content in this paragraph 5. Table 2 is the sampling criteria. The findings does not reflect views across the criteria used for sampling. Did the views of HCWs differ across the sampled and years of experience? 6. Line 149-155 Data analysis was performed by RP. Could you explain how biasness was addressed. In qualitative research, it is recommended that at least 2 to 3 researchers read and code the transcripts. This is a major weakness in your data analysis. Could the authors explain how they transitioned from coding to the themes. Please add codes, categories and themes table as an appendix. 7. Line 157-161 -add ethics reference number. 8. Findings Authors to consider revisiting the data to improve the findings section, which is weak. To improve this section: - Add quotes from the participants, comparing participants views across years of service and the location of the facilities. Themes are interesting but are not adequately supported by data. For example,
---

	line 189, you state that many Health care workers XXX yet you only give one example, which is not sufficient for the reader to understand what you are talking about. Similar patterns are observed in all your themes. 9. Line 366-373 - should be summarized in the background. **Explain consenting process. **More details on interview process required Overall - This is a good study. Data analysis could be revisited to add depth to this paper. Discussion sections could be shortened to create space for more info in the findings sections. Accept with major revision
--	---

VERSION 1 – AUTHOR RESPONSE

Reviewer: 1
Reviewer Name
Ibrahim Kabo Alhassan

Institution and Country
Palladium International Limited, Integrated Health Program (IHP)
Nigeria

Please state any competing interests or state ‘None declared’:
None declare

Please leave your comments for the authors below
General Comments

I commend the authors for the good work they have done. The study has identified critical issues and gaps to provision of timely and adequate referral for women with emergency obstetric complications in rural Sierra Leone. The topic is relevant and addresses an important problem that faces not only Sierra Leone but also other developing countries.

Reply: Thank you kindly for your commendation.

Section specific remarks
Title: Clearly described the article
Abstract:

Objectives: It would be better if the objectives explain the purpose of conducting the study beyond just describing the perspectives of health workers. I would suggest for consideration the objective to be “Investigate factors affecting timely and adequate referral of women with obstetric complications in rural areas of Sierra Leone”

Reply: Thank you for this important comment. We have changed the objectives in the abstract as well as in the background section to better explain the purpose of our study in line with your suggestion.

Results: Line 41. Furthermore, decision-making of I suggest to rephrase the sentence for clarity

Reply: We appreciate your suggestion and have rephrased the sentence for clarity.

ARTICLE SUMMARY Line 51

Reply: Clarified.

Strengths and limitations of this study

Other limitations of the study:

The study did not consider getting perspective of women and relatives on “women’s compliance to referral”

Reply: Thank you for your suggestion; reviewer #2 also mentioned this as a limitation. We agree that this is a limitation of our study and have added it to the relevant sections.

Background: A brief information that describes the current referral system in Sierra Leone could be of added value. Why the choice of Tonkolili District in Sierra Leone?

Reply: A short description of the current referral system in Sierra Leone was added to the Methods section, lines 120-123 (line numbers correspond to the revised version ‘Main Document’). The reason Tonkolili District was chosen was also added in lines 124-128. We added an extra sentence to clarify our reason.

Data Collection Line 123

Data collection section as described in the manuscript should be moved to Study Design and Setting section line 95

Reply: Thank you. In line with your suggestion, we have merged the data collection section with the study design and setting section. Additionally, we have added an additional phrase to line 103 to further clarify the study design.

In addition, under the methodology

1. There is no mention of research subjects – only engagement with district health management team was mentioned.

Reply: You are correct that under the heading “Study subject and public involvement” the research subjects were not mentioned initially. We have now expanded this section to include them in lines 158-159.

2. Authors may consider listening the inclusion and exclusion criteria since is a purposive sampling

Reply: In line with your suggestion, we have now mentioned and explained the purposive sampling in more detail by providing a list of the selection criteria in lines 132-134.

3. The authors mentioned that a homogenous sampling was used for the health workers without explaining how it was done.

Reply: The reviewer is correct in noting that indeed homogenous sampling was not explained precisely enough. We added a sentence to this extent (also in response to the same point made by Reviewer #2).

The Data Collection Section should basically describe how the data was collected as per design, needs to adjust the data collection instruments, challenges encountered etc

Reply: Thank you. Data collection as per design is now described in lines 129-131. Challenges encountered during data collection is now described in more detail in lines 152-155.

Line 136 Table #3 under competencies of “Midwife 2”. I would be of added value to list the remaining 3 signal functions and whether she could perform them or not

Reply: Thank you for this relevant suggestion. We have edited Table 3 to list the remaining signal functions.

Results:

The Authors would have included the major/common reasons for referring women to higher level health facility

Reply: We agree that a description of the major reasons for referral would have been of added value to our manuscript. Unfortunately, we have not collected this data that reflect these reasons and have included this as a limitation in the discussion section.

Line 232 “Women’s compliance with referral”

The Authors would have considered getting perspective of women and relatives on to balance/validate health workers” perspective

Reply: Our results of the perspectives of health workers could have been validated by the perspectives of women and relatives. However, planning and funding did not allow for additional interviews. Throughout our Discussion section, and especially in lines 377-381, results of previously conducted research including the perspectives of women and relatives are described, which may provide some balance to our results regarding perspectives of health workers. We have also included a notion on the absence of interviews with women and relatives now.

Strengths and Limitations

Another limitation is that the study did not consider getting perspective of women and relatives on “women’s compliance to referral”

Reply: Thank you for your suggestion. This has been added to the relevant sections in the manuscript.

Conclusion

The claims are supported by the results.

Other interventions that may reduce delay in access to adequate emergency obstetric care can based on the findings also include competency based trainings, use of standardized management protocols at different levels, improving referral system with 2-way referral, clinical mentoring.

Reply: Thank you for confirming that our claims are supported by the results. We have added additional interventions along with your important suggested additions.

Reviewer: 2

Reviewer Name

Antonia Arnaert

Institution and Country

McGill University, Ingram School of Nursing, Canada

Please state any competing interests or state ‘None declared’:

None declared

Please leave your comments for the authors below

Paper: Perceptions of health workers on the referral of women with obstetric complications: A qualitative study in rural Sierra Leone.

This is an interesting topic to explore; however, the paper needs

Abstract:

• In the title the authors are talking about “perceptions” and in the abstract they indicate “perspectives”?

Reply: We have changed the manuscript to consistently use the term “perspectives”.

• What is “systematic text condensation”?

Reply: Thank you for noting that this requires additional explanation. Systematic text condensation is now explained in greater detail in the Methods section, lines 162-173.

• More information should be provided on those 19 health workers. Nurses? Midwives?

Reply: Thank you for this suggestion. We have added additional information on the study subjects in the abstract. More detailed information on the study subjects is now provided in the Methods section, lines 136-143 and in Table 3.

- **The results indicate 4 themes but what is the relationships between those themes and the factors that complicate timely and adequate referral?**

Reply: Thank you for this suggestion. We have expanded this now in the Abstract's results section in lines 37-40 to clarify the relationship between these four themes and the factors that complicate timely and adequate referral.

- **The abstract should be improved**

Reply: The abstract has been revised and improved in several sections.

Strengths and Limitations

- **Is it a limitation that the interviews were not triangulated with quantitative data? Why?**

Reply: Reviewer #1 has also mentioned this as a limitation of our study and we have included this now as a limitation in our manuscript.

- **A limited number of participants? The authors interviewed 19 participants. Why is this a limitation?**

Reply: We agree that with 19 participants, we have reached data saturation. Additional interviews and participants were also not possible in our current study design.

Background

- **No literature available? What about evidence in other countries?**

Reply: No data concerning the perspectives of health workers in rural Sierra Leone is available. However, there is data from other countries in sub-Saharan Africa which describe the perspectives of health workers concerning the referral of women with obstetric complications. These studies have been extensively compared with and referred to in the Discussion section.

Method

- **Why is it important to say that the District Health Management Team was involved in the design, conduct, reporting and dissemination planning of our research?**

Reply: As part of its patient and public partnership strategy, BMJ Open encourages active patient and public involvement in clinical research. In line with this, we felt it important to state that there was active public involvement in our study.

- **In the paper it says 9 semi-structured interviews (instead of 19). What is the number of participants?**

Reply: The number of participants in the interviews was not sufficiently clear in the manuscript. A respective sentence has now been added in lines 139-140 in the Methods section to clarify the number of participants.

- **Is the Master of Medicine student a co-author? Perhaps indicate that the interview by 1st or ... co-author?**

Reply: Thank you for this suggestion. We have clarified this.

- **How were those facilities selected? Based on the number of high-risk pregnancies? What is the % of at-risk mothers in those districts?**

Reply: The reasons the three chiefdoms in Northern Tonkolili District were selected as study area are described in the first part of the Study design and setting section, lines 124-128. Purposive sampling, described in lines 131-134, was used to select the facilities where the interviews were held. Quantitative data concerning the % of at-risk mothers in these districts is shortly described in the Background section, lines 78-83. Approximately 50% of women in rural areas of Sierra Leone give birth outside a health facility.

• A purposive sampling was used to select facilities and a homogenous sampling was used to select healthcare workers? Unclear/

Reply: Thank you for this comment. This was a two-step approach. First, a sample of health facilities where interviews were conducted was taken. It is described in the Study design and setting section in lines 131-134 how variation between selected facilities was ensured by purposive sampling. Second, we used a sample of health workers participating in the interview in the already selected health facility. In lines 136-139 it is described how health workers with different health cadres but working at the same health facility were selected using homogenous sampling. We have clarified this, also in line with reviewer #1.

• 19 Healthcare workers participated in 9 interviews? Why? How was this done? Focus groups?

Reply: First, nine facilities were selected using purposive sampling as described as mentioned above, and in lines 131-134 and Table 2. Second, all health workers working at the selected facilities were invited to participate in the interviews. This resulted in some interviews being conducted with only one health worker, and some interviews with up to four health workers. The total number of health workers participating in the nine interviews was nineteen. This has been further clarified in the manuscript.

• How were those healthcare workers selected?

Reply: This comment has been addressed extensively, as explained in the above comments.

• In the table it says: “health worker (number interviewed)”. No number is provided.

Reply: In Table 3, each row describes a different health cadre, followed in brackets by the number of health workers of that health cadre participating in the interviews, and a short description of their competencies.

• What about trustworthiness of data?

Reply: Thank you for raising this concern. Reviewer #3 also addresses this topic. All steps of data analysis have been double-checked by the second author in order to guarantee trustworthiness of data as much as possible. All authors declare they have no competing interests.

Results

• There is no introduction to the results?

Reply: Thank you for this comment. To further introduce our results, we have added a sentence before describing the four resulting themes.

• It would be good to describe the roles of those health workers in case of a referral.

Reply: Throughout the results section, we describe what decisions and actions a health worker must consider throughout the referral process. For example, information on history-taking from women in lines 245-246, asking for advice regarding the decision to refer in lines 189-191, dealing with relatives in lines 252-260, and the clinical management of a woman while waiting for transport in lines 209-212 are provided.

• The nuances of those different health workers must be better described in the results.

Reply: Thank you for this comment. After each quote, a short description of the participant is given including the gender, the health worker cadre, and the level of the facility where this health worker works. The age and years of experience of the participants is not known to us.

• **Quotes are not rich descriptions and not always meaningful to the text.**

Reply: We agree that this may have been the case and thank you very much for noting this. In line with you and Reviewer #3, we have provided additional quotes in the revised manuscript.

• **Why using references in results? Under logistical constraints**

Reply: We have omitted the references from the results section.

Discussion: No discussion provided

Reply: An extensive discussion section has been provided in lines 327-427.

Reviewer: 3
Reviewer Name
Adelaide Lusambili

Institution and Country
Aga Khan University
Kenya

Please state any competing interests or state 'None declared':
N/A

Please leave your comments for the authors below

1. Line 2: COREQ - spell this acronym and all other acronyms in the paper first time.

Reply: Thank you for this suggestion. We have spelled out the acronyms in the paper the first time they are used.

2. Abstract

Remove HCWs from settings and leave it under the heading "participants".

Reply: We have amended the abstract as suggested.

3. Line 33-60. Remove bullet points. Write as a paragraph. You have the same information from line 374. I suggest, you delete this information from line 33 -60 and move to line 374.

Reply: The strengths and limitations in lines 57-66 are given in bullet points and written as a paragraph in lines 412-427 to comply with the submission guidelines of BMJ Open (<https://bmjopen.bmj.com/pages/authors/>).

4. Line 120, remove heading and merge line 121-122 with the content in this paragraph

Reply: In line with your suggestion, we have merged the data collection section with the study design and setting section.

5. Table 2 is the sampling criteria. The findings does not reflect views across the criteria used for sampling. Did the views of HCWs differ across the sampled and years of experience?

Reply: Thank you for this comment. We feel that our selection of facilities and health workers accurately represents all chiefdoms and different cadres and characteristics of professionals.

6. Line 149-155

Data analysis was performed by RP. Could you explain how biasness was addressed. In qualitative research, it is recommended that at least 2 to 3 researchers read and code the transcripts. This is a major weakness in your data analysis.

Reply: Thank you for this valuable comment. In our original manuscript, the role of HM, the second author, in the data analysis was not clearly described. We have expanded the analysis section to clarify the role of HM, and to thereby highlight the trustworthiness of our data and the measures we have taken to address possible bias in our results. She was fully involved in reading and assisting with coding of the transcripts.

Could the authors explain how they transitioned from coding to the themes. Please add codes, categories and themes table as an appendix.

Reply: We have now added an extensive description of the process of data analysis through systematic text condensation as described by Malterud. Additionally, a reference to the original article by Malterud has been added so that interested readers can read the full description of this method.

7. Line 157-161 -add ethics reference number.

Reply: We have official ethical approval of the Sierra Leone Ethics and Scientific Review Committee, which does not provide reference numbers. This issue has already been discussed by the authors with the handling editor of BMJ Open.

8. Findings

Authors to consider revisiting the data to improve the findings section, which is weak. To improve this section:

- Add quotes from the participants, comparing participants views across years of service and the location of the facilities. Themes are interesting but are not adequately supported by data. For example, line 189, you state that many Health care workers XXX yet you only give one example, which is not sufficient for the reader to understand what you are talking about. Similar patterns are observed in all your themes.

Reply: Thank you kindly for this remark. Based on your suggestion, we have revisited the data and considerably expanded and improved the findings section. We have added multiple quotes, in order to adequately support the themes by our data. In the revised manuscript, 15 quotes are included to illustrate our findings.

However, we cannot compare the participants views across years of service as years of service were unknown to us. Also, we cannot compare across the location of the facilities as this is not the aim of our study. As mentioned in a previous comment, the selection of the location of the facilities was meant to accurately represent the whole study area and not to identify differences within the study area.

9. Line 366-373 - should be summarized in the background.

Reply: In line with your suggestion, we have now summarized this information in the Background section.

****Explain consenting process.**

Reply: The consenting process of facilities where interviews were conducted and of health workers who participated in the interviews was better explained in lines 130-131, lines 134-135, and in lines 160-161. Written informed consent was obtained from interview participants.

****More details on interview process required**

Reply: The interview process is now described in greater detail in lines 144-155. The use of an interview guide is extensively described in the manuscript.

Overall - This is a good study. Data analysis could be revisited to add depth to this paper. Discussion sections could be shortened to create space for more info in the findings sections.

Reply: Thank you kindly for your positive and valuable feedback and comments. We have revisited our description of the data analysis to add depth to the paper. Also, we have added quotes in the results section to further support our claims. We thank you for your peer-review, which greatly helped to improve our paper.

Accept with major revision

VERSION 2 – REVIEW

REVIEWER	Ibrahim Alhassan Kabo Palladium International Ltd/Integrated Health Program, Nigeria
REVIEW RETURNED	21-Oct-2020
GENERAL COMMENTS	General Comments I commend the authors for the good work they have done in revising the manuscript submitted earlier. The incorporation of comments has greatly improved the quality of the manuscript. Section specific remarks All previous comments and suggestions made in the different sections of the manuscript (abstract, article summary, background, data collection, methodology, results, strengths and limitations and conclusion) were incorporated as appropriate.